# Investigating genetic links of vitamin D metabolism pathway genes (*CYP2R1, CYP27B1, CYP24A1*, and *DBP)* in Multiple Sclerosis patients

Laith AL-Eitan[1,2]*, Asaad Ataa[1]

1 Department of Applied Biological Sciences, Jordan University of Science and Technology, Irbid, Jordan,
2 Department of Biotechnology and Genetic Engineering, Jordan University of Science and Technology, Irbid, Jordan

* lneitan@just.edu.jo

## Abstract

Multiple sclerosis (MS) is believed to result from a complex interplay of behavioral, genetic, and environmental risk factors. Furthermore, some studies indicated that vitamin D deficiency is linked to the emergence of different diseases, including MS. This study aims to determine the genetic associations between vitamin D metabolism gene polymorphisms and MS susceptibility in the Jordanian community. A total of 388 samples (192 MS patients and 196 controls). Genotypes for *CYP2R1* (rs10741657, rs12794714), *CYP27B1* (rs10877012), *CYP24A1* (rs2248359), and *DBP* (rs7041, rs4588) were determined by PCR/RFLP assay method. The study revealed a significant association with increased MS risk in SNPs rs10877012 (C/A, *P=0.0002*) of the *CYP27B1* gene and rs4588 (A/A, *P=0.04*) of the *DBP* gene. Additionally, the haplotypes of the *CYP2R1* gene revealed a significant association with MS patients and controls (GG, p=1e-04; AA, *p<0.0001*). Moreover, only a SNP rs4588 of the *DBP* gene has been significantly associated (*P=*0.04) with a clinical phenotype of multiple sclerosis and vitamin D deficiency. Understanding these genetic variations in multiple sclerosis susceptibility genes can help healthcare professionals improve early diagnosis and develop personalized treatment options.

## 1. Introduction

Multiple sclerosis (MS) is a chronic disease that targets the central nervous system (CNS) [1]. The disease is diagnosed based on clinical findings and supporting proof from ancillary exams such as brain magnetic resonance imaging (MRI) and cerebrospinal fluid (CSF) examinations [2,3]. MS is classified as a complex genetic disorder, common in populations with moderate genetic risk factors and involving intricate interactions between genetics and environment [4]. It often affects a specific age group, typically between 20 and 45 years old [5,6]. The disease commonly begins around age 20 and tends to worsen with age and disease duration [6]. Globally, the

**Data availability statement:** All relevant data are within the paper and its Supporting Information files.

**Funding:** Prof. Laith AL-Eitan received a fund from the Deanship of Research at Jordan University of Science and Technology, grant No. 15/2020. The funders had no role in study design, data collection and analysis, decision to publish, or preparation of the manuscript.

**Competing interests:** The authors have declared that no competing interests exist.

prevalence ranges from 50 to 300 cases per 100, 100,000 people, with over two million cases reported worldwide.

Additionally, the average age of MS onset is younger than that of healthy individuals [7]. Vitamin D plays a vital role in immune regulation and neuroprotection. There are two forms of vitamin D: vitamin D 3 (Cholecalciferol) and vitamin D 2 (Ergocalciferol). Vitamin D3 is produced naturally in the skin through exposure to ultraviolet B (UVB) rays from the sun, while vitamin D2 is synthesized from ergosterol [8,9]. Vit D3 is generated endogenously after sunlight exposure, and vitamin D is also obtained through diet [10]. Although this form of vitamin D is inactive initially, it undergoes hydroxylation at position C-25 in the liver, carried out by several enzymes involved in vitamin D hydroxylation, including CYP2R1 and CYP27A1. The kidney enzyme CYP27B1 then converts it at carbon 1, producing the active hormonal form of vitamin D. This enzyme is also found in other tissues such as bone cells, skin, colon, brain, and macrophages. Variations in this process can significantly impact vitamin D levels [11–13]. Vitamin D deficiency is among the most widespread health issues globally. A major cause of this deficiency is insufficient sun exposure, as sunlight is the primary source of vitamin D for most people. Few foods naturally contain vitamin D, and dietary fortification often falls short of meeting the needs of children and adults. Vitamin D deficiency can have long-term effects, including rickets in children and osteomalacia and osteoporosis in adults. Furthermore, vitamin D deficiency has been linked to increased risks of common cancers, autoimmune diseases, hypertension, and infectious diseases [14,15].

Additionally, serious adverse effects, including a higher risk of MS, colon, prostate, breast, and ovarian cancers, as well as type 1 diabetes, have been linked to low vitamin D levels in recent years [15,16]. Genetic and functional studies of vitamin D pathways have shown a connection between MS and other diseases with vitamin D deficiency [17–19]. Some studies also suggest that vitamin D deficiency may influence the development of type 1 diabetes, asthma, colon cancer, multiple sclerosis, and chronic obstructive pulmonary disease by examining genes involved in the vitamin D pathway [18,20–22]. Therefore, a common explanation for the impact of latitude on MS is the idea that sun exposure protects against the disease by increasing vitamin D3 levels [23], and that deficiency might raise the risk of MS [4]. UV radiation is the primary trigger for vitamin D3 synthesis in humans [4].

This suggests that the activity of vitamin D3 is influenced by environmental factors, including exposure to sunlight and consumption of foods rich in vitamin D [10,24]. Recent genetic studies have shown that various common risk Single Nucleotide Polymorphisms (SNPs) associated with multiple sclerosis are located within or near genes involved in complex vitamin D (Vit D) metabolism [25]. Furthermore, several studies have focused on the *CYP1R2*, *CYP27B1*, and *DBP* genes and their association with diseases such as multiple sclerosis. Additionally, multiple studies have examined various genes, including *CYP1R2*, *CYP27B1*, *CYP24A1*, and *DBP*, and their connection to the development of diseases such as multiple sclerosis.

This research is the first study to examine the associations between SNPs rs10741657, rs12794714 of *CYP2R1*, rs10877012 of *CYP27B1*, rs2248359 of

*CYP24A*, and rs7041, rs4588 of the *DBP* gene polymorphisms with MS risk in Jordan. The study focuses on highlighting the two main causes of MS—genetics and environmental factors—by investigating vitamin D as a genetic component through polymorphisms in vitamin D metabolic genes and as an environmental factor by measuring serum vitamin D levels. This approach aims to provide healthcare professionals with better early diagnosis and more treatment options for MS patients.

## 2. Subjects and methods

### 2.1 Subjects

A total of 388 participants were enrolled in the study, comprising 192 patients with a clinical diagnosis of multiple sclerosis (MS) and 196 controls without MS. Among the controls, some individuals presented with vitamin D deficiency. We conducted this study in the Biotechnology & Genetic Engineering laboratory at Jordan University of Science & Technology (JUST) in Jordan. The average age of the participants was 36 years at the time the blood sample was taken.

Additionally, Prof. AL-Eitan provided DNA samples from patients and controls as part of his research on the genetic associations of multiple sclerosis (MS) in Jordanian patients. The study received approval from the Institutional Review Board (IRB) at Jordan University of Science and Technology (JUST) under the ethical code (12/108/2017). The collection of samples from all involved institutions—Princess Basma Hospital, King Abdullah University Hospital, the Jordanian Royal Medical Services, and Al-Basheer Hospital—was conducted under the same ethical approval (Approval #: 12/108/2017, dated August 29, 2017) granted by the IRB at JUST, ensuring consistency in adherence to the guidelines, and occurred from January 1, 2018, to January 1, 2019 Before collecting blood samples, a thorough inclusion/exclusion criterion was applied. First, participants needed to have an officially confirmed diagnosis of multiple sclerosis. The following characteristics were considered exclusion criteria: first or second-degree relatives of the participating patient in this study, patients with unclear or incomplete clinical data, patients who did not provide written consent to participate, or patients who have a phobia of needles or blood. All participants provided written informed consent before their inclusion in the study. Furthermore, the IRB at JUST granted additional ethical approval (Approval #: 39/128/2019, dated 28/11/2019) for a follow-up investigation on the genetic associations of MS in Jordanian patients, covering the period from January 2, 2020, to December 4, 2020.

### 2.2 SNP selection and genotyping

According to the literature, multiple gene polymorphisms may be associated with the development and susceptibility to MS. This has influenced the selection of genes and candidate polymorphisms in this study. In the current study, SNPs rs10741657 [26], rs12794714 [27] of *CYP2R1*, rs10877012 [28] of *CYP27B1, and* rs2248359 [29] of *CYP24A,* as well as rs7041 and rs4588 of *DBP* [30], *were investigated* due to reports of their association with MS development in multiple populations.

According to the manufacturer's instructions, a commercial kit (Qiagen, Germany) was used to extract genomic DNA from blood samples. Additionally, the purity and concentration of the extracted DNA were assessed using a BioDrop µLite spectrophotometer (England). The primer sequences for the SNPs (rs10741657, rs12794714, rs10877012, rs228359, rs7041, and rs4588) of the *CYP2R1, CYP27B, CYP24A1*, and *DBP* genes are listed in Supplementary S1 Table, along with the restriction enzymes used in our study. Moreover, the work environment was adjusted and continuously sterilized during each PCR operation to ensure that no DNA contamination occurred.

On the other hand, samples of genotypes already known to have been subjected to RFLP, with each process running as a positive control sample, were used to ensure that the enzymes cut off PCR products correctly.

An analysis was performed to detect polymorphisms. Firstly, the desired region of the gene was copied by PCR, followed by RFLP. Then, a 3% agarose gel was used to separate 10 µL of each digested PCR product and 5 µL of 50 bp

and 100 bp ladders. A 5 µL of 10 mg/mL ethidium bromide was added to the gel for staining the bands. Supplementary S2 Table shows the DNA's PCR product size and the SNPs' restriction fragment sizes. Mentioning the appropriate temperatures after performing thermal profile optimization of the polymerase chain reactions of each SNP. Meanwhile, the Supplementary figures (S1, S2, S3, S4, S5, and S6 Figs) show all SNPs' representative gels of PCR-RFLP products.

Quality control measures were implemented to ensure the reliability of genotyping data. SNPs with call rates below 95% and samples with call rates below 90% were excluded from the analysis. Genotyping accuracy was evaluated using duplicate samples, with an observed error rate of less than 1%. Ambiguous genotypes—defined as samples with unclear or non-distinct banding patterns on gel electrophoresis—were excluded from the analysis to reduce the risk of misclassification.

### 2.3  Clinical and biochemical test (Measurement of vitamin D levels)

Serum 25(OH)D was measured for cases and healthy controls by MAGLUMI 25(OH)D (CLIA) kit, which is an in vitro chemiluminescence immunoassay for the quantitative determination of 25(OH)D in human serum using MAGLUMI series fully-auto CLIA analyzer (MAGLUMI 800) in the Biotechnology and Genetic Engineering Laboratory at JUST.

According to the American Association of Clinical Endocrinologists, individuals with a vitamin D level below 30 ng/ml are considered to have a deficiency. In comparison, individuals with 30–50 ng/ml have optimal (sufficiency) vitamin D [31].

### 2.4  Statistical analysis

The Statistical Package for Social Science performed the data analyses for SPSS version 17. The web tool SNPStats (https:/snpstats.net/start.htm) and the Chi-square test were used to evaluate the risk association of alleles and genotype frequencies. The genotype-phenotype correlation for the 6 SNPs among MS patients was tested using analysis of variance (ANOVA) or Pearson's Chi-square test with a p-value of less than 0.05. The Hardy-Weinberg equilibrium (HWE) was calculated, and statistical significance was defined as a P-value less than 0.05. A multiple comparison adjustment was applied using the Bonferroni correction to minimize Type I error. The significance threshold was set at $\alpha = 0.05$ divided by the number of independent comparisons.

## 3.  Results

### 3.1  Descriptive analysis

The current case/control study involved 192 unrelated Jordanian patients diagnosed with multiple sclerosis. These patients were chosen randomly during their follow-up visits to hospital neurology clinics. Patients included both males and females; the average age was $36.17 \pm 10.37$ years, with a median of 36 and a range of 15.5–64. A 196 group of healthy donors was chosen as a control group with an average age of $26.94 \pm 6.15$, and a median of 25, and the range was (20–39) years. A matched gender of patients was selected with (29.17%) males and (70.83%) females, while matched gender healthy controls were selected with (31.63%) males and (68.37%) females. Moreover, all the participants fulfilled the inclusion criteria and agreed to contribute to this study. The patient description, including demographic and clinical data, as well as the demographic data for healthy controls, is summarized in Table 1.

### 3.2  Hardy-Weinberg Equilibrium (HWE)

After performing the HWE test to assess polymorphisms in both case and control subject sample sets, all SNPs met the HWE standards in both groups and were included in this study. The Hardy-Weinberg equilibrium of the included SNPs is shown in Table 2. The test indicates that SNP rs10741657 (CYP2R1) and rs2248359 (CYP24A1) were the only 2 SNPs normally distributed among both cases and controls. A possible explanation for this is the prevalence of consanguineous marriages in the Jordanian population, which may lead to genetic isolation and homogeneity [32].

**Table 1. Description of demographic and clinical characteristics of MS patients and healthy controls.**

| Characteristic | Mean ± SD with % | |
|---|---|---|
| | **Patient** | **Control** |
| Number | 192 | 196 |
| Male | 56 (29.17%) | 62 (31.63%) |
| Female | 136 (70.83%) | 134 (68.37%) |
| Age (years) | 36.17 ± 10.37 | 26.94 ± 6.15 |
| BMI (kg/m2) | 24.71 ± 5.33 | 24.84 ± 4.1 |
| Clinical data | | |
| Vitamin D Deficiency | | |
| Yes | 92.40% | 37.24% |
| No | 7.60% | 62.76% |

**Table 2. CYP2R1, CYP27B1, CYP24A1, and *DBP* (GC) SNPs, their minor allele frequencies, and HWE p-value in cases and controls.**

| Gene | SNP _ID | Cases (n = 192) | | | Controls (n = 196) | | |
|---|---|---|---|---|---|---|---|
| | | MA[a] | HWE[c]*p-value* | MAF[b] | MA[a] | MAF[b] | HWE[c]p-value |
| *CYP2R1* | rs10741657 | A | 0.7 | 26% | A | 27% | 0.28 |
| | rs12794714 | G | 0.47 | 46% | G | 49% | 0.023 ᐃ |
| *CYP27B1* | rs10877012 | A | <0.0001ᐃ | 28% | A | 27% | <0.0001 ᐃ |
| *CYP24A1* | rs2248359 | T | 0.89 | 49% | T | 43% | 0.25 |
| *DBP* | rs7041 | T | 0.77 | 43% | T | 41% | 0.038 ᐃ |
| | rs4588 | A | 0.49 | 19% | A | 21% | 0.0075 ᐃ |

[a]MA: Minor allele.

[b]MAF: Minor allele frequency.

[c]HWE: Hardy-Weinberg equilibrium.

ᐃSignificant P values are indicated by a triangle (ᐃ P < 0.05).

### 3.3 Association of SNPs Candidate Genes with Multiple Sclerosis

The allele and genotype frequencies of SNP rs10877012 C/A in 5' near gene 12q14.1 of the *CYP27B1* gene in chromosome 12 show the association of the C/A genotype with the risk of MS. Accordingly, the genotype frequency in the MS cases group was significantly different from the controls group, the overall estimate of effects in OR 0.05 ($\chi$2 (2, N = 388) =17.27, *P* = 0.0002) where the C/A genotype was found to be less frequent among MS cases (1%) compared to the controls (10%). No significant differences in allele frequencies were found (*P = 0.73*), with the A allele being more frequent among MS cases (28%) compared to the controls (27%). Meanwhile, the C/allele was found to be more frequent among controls (73%) than in MS cases (72%).

Moreover, the allele and genotype frequencies of SNP rs4588 C/A in exon 11 of the *DBP* gene showed an association between the A/A genotype and the risk of MS. Accordingly, the genotype frequency in MS cases was significantly different from that of the controls, the overall estimate of effects in OR 0.35 ($\chi$2 (2, N = 388) = 6.13, *P* = 0.046). Additionally, the A/A genotype was less frequent among MS cases (3%) compared to the controls (8%). On the other hand, no significant differences in allele frequencies were found (*P = 0.50*), with the C allele being more frequent among MS cases (81%) compared to the controls (79%). Additionally, the A/allele was found to be more frequent among controls (21%) compared to MS cases (19%), as shown in Table 3. After applying the Bonferroni correction, a significant association was observed only between the rs10877012 SNP of the *CYP27B1* gene and the disease.

**Table 3. Association of genes SNPs with multiple sclerosis susceptibility.**

| Gene | SNP_ID | Allele/ Genotype | Cases (192) | Controls (196) | Cases vs. Controls | |
|------|--------|------------------|-------------|----------------|-----|-----|
| | | | | | Chi-squared | P -value |
| *CYP2R1* | rs10741657 | G | 286 (74%) | 286 (73%) | 0.23 | 0.63 |
| | | A | 98 (26%) | 106 (27%) | | |
| | | GG | 105 (55%) | 101 (51%) | 0.44 | 0.803 |
| | | GA | 76 (39%) | 84 (43%) | | |
| | | AA | 11 (6%) | 11 (6%) | | |
| | rs1279714 | A | 209 (54%) | 200 (51%) | 0.9 | 0.341 |
| | | G | 175 (46%) | 192 (49%) | | |
| | | AA | 54 (28%) | 59 (30%) | 5.68 | 0.058 |
| | | AG | 101 (53%) | 82 (42%) | | |
| | | GG | 37 (19%) | 55 (28%) | | |
| *CYP27B1* | rs10877012 | C | 275 (72%) | 285 (73%) | 0.11 | 0.734 |
| | | A | 109 (28%) | 107 (27%) | | |
| | | CC | 137 (71%) | 133 (68%) | 17.27 | **0.0002 ᐃ** |
| | | CA | 1 (1%) | 19 (10%) | | |
| | | AA | 54 (28%) | 44 (22%) | | |
| *CYP24A1* | rs2248359 | C | 196 (51%) | 222 (57%) | 2.66 | 0.102 |
| | | T | 188 (49%) | 170 (43%) | | |
| | | CC | 49 (26%) | 67 (34%) | 3.47 | 0.17 |
| | | CT | 98 (51%) | 88 (45%) | | |
| | | TT | 45 (23%) | 41 (21%) | | |
| *DBP* | rs7041 | G | 220 (57%) | 232 (59%) | 0.29 | 0.593 |
| | | T | 164 (43%) | 160 (41%) | | |
| | | GG | 64 (33%) | 76 (39%) | 2.04 | 0.361 |
| | | GT | 92 (48%) | 80 (41%) | | |
| | | TT | 36 (19%) | 40 (20%) | | |
| | rs4588 | C | 312 (81%) | 311 (79%) | 0.45 | 0.503 |
| | | A | 72 (19%) | 81 (21%) | | |
| | | CC | 125 (65%) | 130 (66%) | 6.13 | 0.046 ᐃ |
| | | CA | 62 (32%) | 51 (26%) | | |
| | | AA | 5 (3%) | 15 (8%) | | |

ᐃSignificant P values are indicated by a triangle (ᐃ P<0.05).

P-values<0.0041 (0.05/12, after Bonferroni correction for multiple comparisons) were considered statistically significant, and are indicated in bold.

### 3.4 Genetic association of CYP2R1, CYP27B1, *CYP24A1*, and DBP Polymorphisms with multiple sclerosis for the genetic models

In the SNP rs12794714, no genetic associations were found in either the co-dominant or dominant genetic models. At the same time, the common homozygous (GG) and heterozygous (AG) versus rare homozygous (AA) were found to be significantly associated with a risk of MS in the recessive genetic model (OR = 0.61, 95% CI: 0.38–0.98, $P=0.041$). Additionally, the heterozygous (AG) versus the common homozygous (GG) and rare homozygous (AA) genotypes were found to be significantly associated with a risk of MS in the over-dominant genetic model (OR = 1.54, 95% CI: 1.03–2.30, *P=0.034*) of the *CYP2R1* gene. However, for the SNP re10741657 in the *CYP2R1* gene, no genetic associations were found in any of the genetic models with MS.

The co-dominant genetic model was found to be more significantly associated with a risk of MS (OR = 0.05, 95% CI: 0.01–0.39, $P < 0.0001$), where the heterozygous (CA) genotype was less frequent among MS cases (0.5%) compared to the healthy controls (9.7%). Also, the common homozygous (CC) and the heterozygous (CA) versus the rare homozygous (AA) were found to be more significantly associated with a risk of MS in the over-dominant genetic model (OR = 0.05, 95% CI: 0.01–0.37, $P < 0.0001$). Nevertheless, no genetic associations were found within either the dominant or recessive genetic models for SNP rs10877012 of the *CYP27B1* gene. After adjustment for multiple comparisons with the Bonferroni correction, a significant association was observed only for the rs10877012 SNP under the codominant and overdominant models.

For the SNP rs2248359 in the *CYP24A1* gene, no genetic associations were found within all genetic models with MS. Additionally, the co-dominant genetic model was found to be significantly associated with a risk of MS (OR = 0.35, 95%; CI: 0.12–0.98, $P = 0.042$) where the homozygous (AA) genotype was less frequent among MS cases (2.6%) compared to the healthy controls (7.7%). Also, the common homozygous (CC) and the heterozygous (CA) versus the rare homozygous (AA) were found to be significantly associated with a risk of MS in the recessive genetic model (OR = 0.32, 95% CI: 0.11–0.91, $P = 0.022$). However, no genetic associations were found within the dominant and the over-dominant genetic models in SNP rs4588 of the *DBP* gene. For the SNP re7041 in the *DBP* gene, no genetic associations were found within all genetic models with MS, as shown in Table 4.

### 3.5 Haplotype analysis of the *CYP*2R1 gene

The haplotypes of two SNPs on the promoter of the *CYP2R1* gene revealed a significant association with multiple sclerosis patients and controls. The n-Alu repeat polymorphisms of the *CYP2R1* gene (GG, p = 1 × 10^ (−4), and AA, $P < 0.0001$) were associated with an increased risk of multiple sclerosis (Table 5). Following Bonferroni correction, a significant association persisted for the GG and AA haplotypes.

### 3.6 Haplotype analysis of the *DBP* gene

The haplotypes of two SNPs in the promoter region of the *DBP* gene showed no genetic association between multiple sclerosis patients and healthy controls (Table 6). No significant associations persisted after Bonferroni adjustment.

### 3.7 Genotype versus phenotype correlation among MS patients

The results showed that the SNP rs4588 of the *DBP* gene was associated with the vitamin D deficiency variable in MS patients ($P = 0.040$). At the same time, the SNPs (rs1279714, rs2248359, and rs7041) of the *CYP2R1*, *CYP24A1*, and *DBP* genes have been associated with the gender variable in MS patients (*P = 0.043*, *P = 0.038*, and *P = 0.017*), respectively. Furthermore, the results showed that the SNP rs10877012 of the *CYP27B1* gene was associated with the BMI variable in MS patients (P = 0.011), as shown in Table 7. After adjustment for multiple comparisons using the Bonferroni correction, no significant associations were observed between any of the SNPs and clinical characteristics.

## 4. Discussion

Multiple sclerosis (MS) is the most common inflammatory disorder of the brain and spinal cord that mainly develops in young individuals [33]. Additionally, it is an autoimmune disease resulting from a complex interplay between genetic, lifestyle, and environmental risk factors, including both infectious and non-infectious factors [1]. Genetic and functional studies point to a critical role of vitamin D (Vit D) in MS and other diseases linked to a vitamin D deficiency by studying genes involved in the Vit D pathway [17–19]. Vitamin D is a group of fat-soluble Ketosteroid hormones with functional and regulatory effects in the body. It has been involved in the development of the brain and spinal cord. The active form of vitamin D (1,25 (OH) 2D) also has potent anti-inflammatory and immunomodulatory properties [34].

**Table 4. Genetic models and distributions of SNPs within (*CYP2R1, CYP27B1, CYP24A1,* and *DBP*) genes in 192 cases and 196 controls.**

| Gene | SNP ID | Model | Cases (%) | Controls (%) | OR (95% CI) | p-Value * |
|---|---|---|---|---|---|---|
| *CYP2R1* | rs10741657 | G/G | 105 (54.7%) | 101 (51.5%) | 1.00 | 0.8 |
| | | G/A | 76 (39.6%) | 84 (42.9%) | 0.87 (0.58–1.32) | |
| | | A/A | 11 (5.7%) | 11 (5.6%) | 0.96 (0.40–2.32) | |
| | | GG | 105 (54.7%) | 101 (51.5%) | 1.00 | 0.53 |
| | | G/A – A/A | 87 (45.3%) | 95 (48.5%) | 0.88 (0.59–1.31) | |
| | | G/G – G/A | 181 (94.3%) | 185 (94.4%) | 1.00 | 0.96 |
| | | AA | 11 (5.7%) | 11 (5.6%) | 1.02 (0.43–2.42) | |
| | | G/G – A/A | 116 (60.4%) | 112 (57.1%) | 1.00 | 0.51 |
| | | A/G | 76 (39.6%) | 84 (42.9%) | 0.87 (0.58–1.31) | |
| | rs1279714 | A/A | 54 (28.1%) | 59 (30.1%) | 1.00 | 0.058 |
| | | A/G | 101 (52.6%) | 82 (41.8%) | 1.35 (0.84–2.15) | |
| | | G/G | 37 (19.3%) | 55 (28.1%) | 0.74 (0.42–1.28) | |
| | | AA | 54 (28.1%) | 59 (30.1%) | 1.00 | 0.67 |
| | | A/G – GG | 138 (71.9%) | 137 (69.9%) | 1.10 (0.71–1.71) | |
| | | A/A – A/G | 155 (80.7%) | 141 (71.9%) | 1.00 | 0.041 ᐃ |
| | | G/G | 37 (19.3%) | 55 (28.1%) | 0.61 (0.38–0.98) | |
| | | A/A -G/G | 91 (47.4%) | 114 (58.2%) | 1.00 | 0.034 ᐃ |
| | | A/G | 101 (52.6%) | 82 (41.8%) | 1.54 (1.03–2.30) | |
| *CYP27B1* | rs10877012 | C/C | 137 (71.4%) | 133 (67.9%) | 1.00 | **< 0.0001 ᐃ** |
| | | C/A | 1 (0.5%) | 19 (9.7%) | 0.05 (0.01–0.39) | |
| | | A/A | 54 (28.1%) | 44 (22.4%) | 1.19 (0.75–1.90) | |
| | | C/C | 137 (71.3%) | 133 (67.9%) | 1.00 | 0.45 |
| | | C/A – A/A | 55 (28.6%) | 63 (32.1%) | 0.85 (0.55–1.31) | |
| | | C/C – C/A | 138 (71.9%) | 152 (77.5%) | 1.00 | 0.2 |
| | | A/A | 54 (28.1%) | 44 (22.5%) | 1.35 (0.85–2.14) | |
| | | C/C -A/A | 191 (99.5%) | 177 (90.3%) | 1.00 | **< 0.0001 ᐃ** |
| | | C/A | 1 (0.5%) | 19 (9.7%) | 0.05 (0.01–0.37) | |
| *CYP24A1* | rs2248359 | C/C | 49 (25.5%) | 67 (34.2%) | 1.00 | 0.17 |
| | | C/T | 98 (51.1%) | 88 (44.9%) | 1.52 (0.95–2.43) | |
| | | T/T | 45 (23.4%) | 41 (20.9%) | 1.50 (0.86–2.63) | |
| | | C/C | 49 (25.5%) | 67 (34.2%) | 1.00 | 0.062 |
| | | C/T – T/T | 143 (74.5%) | 129 (65.8%) | 1.52 (0.98–2.35) | |
| | | C/C – C/T | 147 (76.6%) | 155 (79.1%) | 1.00 | 0.55 |
| | | T/T | 45 (23.4%) | 41 (20.9%) | 1.16 (0.72–1.87) | |
| | | C/C -T/T | 94 (49%) | 108 (55.1%) | 1.00 | 0.23 |
| | | C/T | 98 (51%) | 88 (44.9%) | 1.28 (0.86–1.91) | |
| *DBP* | rs7041 | G/G | 64 (33.3%) | 76 (38.8%) | 1.00 | 0.36 |
| | | T/G | 92 (47.9%) | 80 (40.8%) | 1.37 (0.87–2.14) | |
| | | T/T | 36 (18.8%) | 40 (20.4%) | 1.07 (0.61–1.87) | |
| | | G/G | 64 (33.3%) | 76 (38.8%) | 1.00 | 0.26 |
| | | T/G – T/T | 128 (66.7%) | 120 (61.2%) | 1.27 (0.84–1.92) | |
| | | G/G -T/G | 156 (81.2%) | 156 (79.6%) | 1.00 | 0.68 |
| | | T/T | 36 (18.8%) | 40 (20.4%) | 0.90 (0.54–1.49) | |
| | | G/G – T/T | 100 (52.1%) | 116 (59.2%) | 1.00 | 0.16 |
| | | T/G | 92 (47.9%) | 80 (40.8%) | 1.33 (0.89–1.99) | |
| | rs4588 | C/C | 125 (65.1%) | 130 (66.3%) | 1.00 | 0.042 ᐃ |
| | | C/A | 62 (32.3%) | 51 (26%) | 1.26 (0.81–1.97) | |
| | | A/A | 5 (2.6%) | 15 (7.7%) | 0.35 (0.12–0.98) | |
| | | C/C | 125 (65.1%) | 130 (66.3%) | 1.00 | 0.8 |
| | | C/A – AA | 67 (34.9%) | 66 (33.7%) | 1.06 (0.69–1.61) | |
| | | C/C – C/A | 187 (97.4%) | 181 (92.3%) | 1.00 | 0.022 ᐃ |
| | | A/A | 5 (2.6%) | 15 (7.7%) | 0.32 (0.11–0.91) | |
| | | C/C – A/A | 130 (67.7%) | 145 (74%) | 1.00 | 0.17 |
| | | C/A | 62 (32.3%) | 51 (26%) | 1.36 (0.87–2.10) | |

ᐃSignificant P values are indicated by a triangle ( ᐃ P < 0.05).

\* Chi-Square test; HWE: Hardy-Weinberg equilibrium test p-value.

P-values < 0.002 (0.05/24, after Bonferroni correction for multiple comparisons) were considered statistically significant, and are indicated in bold.

**Table 5. Frequencies of the *CYP2R1* gene haplotypes among the 192 MS patients and 196 healthy controls.**

| Gene | Haplotypes[a] | Frequency | | Odds Ratio (95% CI) | p-value |
|---|---|---|---|---|---|
| Cases | | | Control | | |
| *CYP2R1* | G A | 0.53 | 0.375 | 1.00 | ------- |
| | G G | 0.214 | 0.354 | 0.48 (0.33–0.68) | **1e-04** [Δ] |
| | A G | 0.241 | 0.135 | 1.20 (0.77–1.85) | 0.42 |
| | A A | 0.0138 | 0.135 | 0.07 (0.02–0.20) | **<0.0001** [Δ] |

[a]Significant P values are indicated by a triangle ([Δ] P < 0.05).

P-values < 0.0125 (0.05/4, after Bonferroni correction for multiple comparisons) were considered statistically significant, and are indicated in bold.

**Table 6. Frequencies of the haplotypes of *DBP* genes among the 192 MS patients and 196 healthy controls.**

| Gene | Haplotypes [a] | Frequency | | Odds Ratio (95% CI) | p-value |
|---|---|---|---|---|---|
| Cases | | | Control | | |
| *DBP* | G C | 0.558 | 0.57 | 1.00 | ------- |
| | T C | 0.253 | 0.223 | 1.15 (0.82–1.60) | 0.42 |
| | T A | 0.173 | 0.185 | 0.96 (0.66–1.40) | 0.84 |
| | G A | 0.014 | 0.021 | 0.68 (0.19–2.39) | 0.54 |

Significant P values are indicated by a triangle ([Δ] P – value).

P-values < 0.0125 (0.05/4, after Bonferroni correction for multiple comparisons) were considered statistically significant, and are indicated in bold.

**Table 7. Association between different CYP2R1, CYP27B1, CYP24A1, and DBP SNPs genotypes and the clinical characteristics of multiple sclerosis.**

| Clinical characteristics | CYP2R1 | | CYP27B1 | CYP24A1 | DBP | |
|---|---|---|---|---|---|---|
| | rs10741657 GG/GA/AA | rs1279714 AA/AG/GG | rs10877012 CC/CA/AA | rs2248359 CC/CT/TT | rs7041 GG/TG/TT | rs4588 CC/CA/AA |
| Gender* | 0.423 | 0.043 [Δ] | 0.746 | 0.038 [Δ] | 0.017 [Δ] | 0.751 |
| Patient age ** | 0.869 | 0.582 | 0.226 | 0.841 | 0.076 | 0.172 |
| Height** | 0.466 | 0.118 | 0.276 | 0.192 | 0.594 | 0.349 |
| Weight** | 0.249 | 0.475 | 0.104 | 0.737 | 0.577 | 0.313 |
| Body mass index ** | 0.086 | 0.106 | 0.011 [Δ] | 0.148 | 0.818 | 0.78 |
| Vit. D deficiency ** | 0.167 | 0.386 | 0.248 | 0.392 | 0.774 | 0.040 [Δ] |

* Pearson's Chi-squared test was used to determine genotype-phenotype association.

** Analysis of variance (ANOVA) test was used to determine genotype-phenotype association.

[a]Significant P values are indicated by a triangle ([Δ] P < 0.05)

P-values < 0.0013 (0.05/36, after Bonferroni correction for multiple comparisons) were considered statistically significant, and are indicated in bold.

Vitamin D exerts its immunomodulatory functions within the immune system by decreasing the presentation of major histocompatibility complex (MHC) class I on T cells and monocytes, thereby reducing T-cell proliferation and the release of pro-inflammatory cytokines [35]. Lower serum vitamin D levels have been reported in MS patients compared to healthy controls. Several studies have reported conflicting results regarding the association between serum vitamin D levels and MS [36].

Several studies have examined the relationship between genetic polymorphisms of vitamin D, including (rs10741657, rs12794714, rs10877012, rs2248359, rs7041, and rs4588), and susceptibility to MS in various countries. However, the results have been inconsistent and controversial [36–41].

Our study confirmed the association of the variant's genotypes between MS patients and healthy subjects using (Co-dominant, Dominant, Recessive, and Over-dominant) genetic models to analyze the genetic association. The associations were found in the recessive and over-dominant genetic models ($P = 0.041$ *and* $P = 0.034$ for the SNP rs1279714 of the *CYP2R1* gene). These findings suggest that the SNP rs1279714 in the *CYP2R1* gene may contribute to the development of MS in the Jordanian population. However, in the SNP rs2248359 of the *CYP24A1* gene, no genetic associations were found within all genetic models with MS. Some studies showed that the involved *CYP2R1* gene is associated with a 2-fold increase in the risk of vitamin D insufficiency and a 38.88% increase in the odds of developing MS for those carrying low-frequency variant causes an increased risk of vitamin D insufficiency in individuals of European descent [42]. Additionally, Laursen and others identified a significant association between 25(OH)D and CYP2R1 (rs10741657, $P = 1.8 \times 10^{-4}$)) in their study, which was conducted in the Danish population [43]. Simon et al. found no association between the *CYP2R1* gene and MS [27].

Furthermore, the co-dominant genetic model was more significantly associated with an increased risk of MS ($P < 0.0001$). Additionally, the association was found in the over-dominant genetic model ($P < 0.0001$) for the SNP rs10877012 of the *CYP27B1* gene. These findings suggest that the SNP rs10877012 in the *CYP27B1* gene may contribute to the development of MS in the Jordanian population.

These results are consistent with a Swedish study by Sundqvist et al. (2010) among the Swedish population with multiple sclerosis, another study by Karaky (2016) on the Spanish population, and a study conducted in Australia and New Zealand by the *ANZ* Gene Consortium on patients with MS. These studies found a relationship between the *CYP27B1* gene and MS susceptibility [10,41,44]. Moreover, the *CYP27B1* gene plays a crucial role in the activation of vitamin D3, and vitamin D3, in turn, plays a vital role in immunological functions. Thus, vitamin D3 is the most likely candidate for susceptibility to MS. However, Orton et al. did not find an association between SNP 10877012 or other SNPs in the *CYP27B1* gene and MS in their study [38].

For the SNP (re2248359) in the *CYP24A1* gene, no genetic associations were found within all genetic models with MS. This result is consistent with a study performed by Simon and others [27] on large cohorts of US nurses among the American population with MS. Sadeghi and others found no association between *CYP24A1* and multiple sclerosis risk in a study performed on the Iranian population [45]. Furthermore, studies conducted on Italian [46] and European [47] populations have found an association between the SNPs (rs2248137 and rs2248359) of the *CYP24A1* gene and susceptibility to MS. Thus, these biological mechanisms require further clarification regarding their association with MS.

For the *DBP* gene, the co-dominant genetic model was significantly associated with an increased risk of MS (*P = 0.042*). The association was also observed in the over-dominant genetic model ($P = 0.022$) for the rs5488 SNP of the *DBP* gene. Therefore, this finding suggests that the rs5488 SNP in the *DBP* gene may contribute to the development of MS in the Jordanian population. These results align with a Danish study by Laursen and colleagues [43] among the Danish individuals with MS, another study by Langer, and a study conducted in Southern California by Kaiser Permanente Southern California, which found that higher 25(OH)D levels were linked with a lower risk of MS in Whites carrying at least one copy of the C allele but not those with the AA genotype [48]. This study included white, non-Hispanic, and Black Hispanic populations. In a meta-analysis that included eight studies, pooled analyses demonstrated no statistically significant associations between the DBP rs7041 and rs4588 polymorphisms and MS risk across genetic models. Subgroup evaluations by ethnicity likewise revealed no significant relationships in either white or non-white populations [49].

Regarding HWE, some SNPs, such as rs10877012 and rs4588, showed slight deviations in the control group. These deviations may result from population-specific factors, such as the high rate of consanguineous marriage, which increases genetic homogeneity within Jordanian society. Some previous studies have indicated that specific SNPs deviating from

HWE may genuinely be associated with disease risk, and excluding them could lead to the loss of true biological associations [50–53]. Therefore, these SNPs were not excluded solely based on their deviation from HWE, as sensitivity analyses were performed and yielded consistent results. Consequently, we included these variants in our analyses to accurately reflect the genetic makeup of the population studied.

The observation that higher serum 25(OH)D levels were associated with a decreased risk of multiple sclerosis in Whites but not in Blacks and Hispanics (Latin American) may be due to differences in allele frequencies at rs4588 and rs7041 among these ethnic groups compared to Whites [54]. A genetic association study was conducted on 94 individuals from 15 families, including at least 2 patients with multiple sclerosis. The analyzed SNPs were located within the *CYP2R1, CYP3A4, CYP27A1, and DBP genes,* and associations between these genes and 25 (OH) D levels were found [55].

Other studies performed on Canadian [56], Italian [57], and Syrian [30] populations did not find an association of the *DBP* gene with MS. Furthermore, we observed a significant interaction between vitamin D deficiency and the rs4588 polymorphism within the *DBP* gene as it relates to the risk of MS with the (A allele) by phenotype-genotype association ($P=0.04$). This result was consistent with a study [58] that reported individuals carrying the rs4588 polymorphism within the DBP gene were more likely to have vitamin D deficiency among healthy Jordanians. Additionally, haplotype analysis indicated that the (GG and AA) haplotypes of the *CYP2R1* gene were significantly associated with a higher risk of multiple sclerosis in the Jordanian population. These results suggest that these genes play a vital role in the development of MS.

The underlying mechanism may involve impaired vitamin D transport, although other factors, such as environmental influences, gender, age, genotypes, smoking, and others, cannot be excluded [27,59]. Finally, some studies have evaluated the association between the vitamin D metabolism polymorphism and the risk of MS. Nevertheless, the results could have been more consistent and conclusive. The reasons may include geographical features, ethnic differences, other environmental and genetic interactions, clinical heterogeneity, or small sample size [60,61].

## 5. Conclusion

This study further supports the gene-environment interaction model of MS. Vitamin D metabolizing genes appear to play an essential role in susceptibility to MS, with the SNPs rs10877012 ($P=0.03$) and rs4588 ($P=0.046$) of the *CYP27B1* and *DBP* genes being associated with an increased risk of MS. Additionally, the haplotype of the *CYP2R1* gene demonstrated a significant association ($P<0.0001$) with the risk of MS. Moreover, a significant correlation was found between the genotype and clinical phenotype of MS vitamin D deficiency.

## 6. Limitations

While this study provides valuable insights into the genetic factors associated with vitamin D deficiency and susceptibility to multiple sclerosis in the Jordanian population, it has some limitations that need to be acknowledged. A limitation of this study is the sample size estimation method, which was based on disease prevalence using the OpenEpi web-based tool. While appropriate for prevalence studies, this approach may not provide sufficient power in a case–control design, potentially contributing to the lack of significant SNP-specific associations. Although the sample size is adequate for preliminary analysis, it remains relatively small, which could limit the generalizability of the results to broader populations or subgroups within Jordan. Larger, multicenter studies are recommended to validate these findings and gain a better understanding of genetic diversity across different regions of the country.

Additionally, the study design, which compares cases with a control group, is inherently susceptible to selection bias, especially since the control group was selected from hospital blood banks and may not fully represent the general healthy Jordanian population. However, we excluded confounding factors such as family history and specific clinical conditions. Variables like sun exposure, dietary habits, and lifestyle factors influencing vitamin D levels were not controlled, which could potentially confound the relationship between genetic variations and vitamin D levels. Vitamin D levels measured only at a single point in time may not accurately reflect long-term vitamin D status or its impact on the development of

multiple sclerosis. Therefore, future longitudinal studies are necessary to establish causality and explore temporal relationships. Finally, given the genetic diversity within Jordan and surrounding populations, larger and more diverse studies are needed to verify these results and investigate additional genetic and environmental factors related to MS.

## Supporting information

**S1 Fig. Supplementary Figure 1 with captions.**
(DOCX)

**S2 Fig. Supplementary Figure 2 with captions.**
(DOCX)

**S3 Fig. Supplementary Figure 3 with captions.**
(DOCX)

**S4 Fig. Supplementary Figure 4 with captions.**
(DOCX)

**S5 Fig. Supplementary Figure 5 with captions.**
(DOCX)

**S6 Fig. Supplementary Figure 6 with captions.**
(DOCX)

**S1 Table. Supplementary Table 1 with SNP primer sequences.**
(DOCX)

**S2 Table. Supplementary Table 2 of PCR product and fragment sizes.**
(DOCX)

**S1 File. Genotype Data Table.**
(XLSX)

## Author contributions

**Conceptualization:** Laith AL-Eitan.

**Data curation:** Laith AL-Eitan, Asaad Ataa.

**Formal analysis:** Laith AL-Eitan, Asaad Ataa.

**Funding acquisition:** Laith AL-Eitan.

**Investigation:** Laith AL-Eitan.

**Methodology:** Laith AL-Eitan, Asaad Ataa.

**Project administration:** Laith AL-Eitan.

**Resources:** Laith AL-Eitan.

**Software:** Laith AL-Eitan.

**Supervision:** Laith AL-Eitan.

**Validation:** Laith AL-Eitan.

**Visualization:** Laith AL-Eitan.

**Writing – original draft:** Laith AL-Eitan, Asaad Ataa.

**Writing – review & editing:** Laith AL-Eitan, Asaad Ataa.

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
