## [Decision Letter · Decision Letter 0]

10 Jul 2025

Dear Dr. Al-Eitan,

Thank you for submitting your manuscript to PLOS ONE. After careful consideration, we feel that it has merit but does not fully meet PLOS ONE’s publication criteria as it currently stands. Therefore, we invite you to submit a revised version of the manuscript that addresses the points raised during the review process.

We look forward to receiving your revised manuscript.

Kind regards,

Wajdy J. Al-Awaida, Ph.D

Academic Editor

PLOS ONE

Journal Requirements:

“Prof. Laith Al-Eitan received a fund from the Deanship of Research at Jordan University of Science and Technology, grant No. 15/2020.”

3. Please note that funding information should not appear in the Acknowledgments section or other areas of your manuscript. We will only publish funding information present in the Funding Statement section of the online submission form. Please remove any funding-related text from the manuscript. 

4. In the online submission form you indicate that your data is not available for proprietary reasons and have provided a contact point for accessing this data. Please note that your current contact point is a co-author on this manuscript. According to our Data Policy, the contact point must not be an author on the manuscript and must be an institutional contact, ideally not an individual. Please revise your data statement to a non-author institutional point of contact, such as a data access or ethics committee, and send this to us via return email. Please also include contact information for the third party organization, and please include the full citation of where the data can be found.

Reviewers' comments:

Reviewer's Responses to Questions

**Comments to the Author**

1. Is the manuscript technically sound, and do the data support the conclusions?

Reviewer #1: Partly

Reviewer #2: Yes

Reviewer #3: Yes

2. Has the statistical analysis been performed appropriately and rigorously?

Reviewer #1: No

Reviewer #2: Yes

Reviewer #3: Yes

3. Have the authors made all data underlying the findings in their manuscript fully available?

Reviewer #1: Yes

Reviewer #2: Yes

Reviewer #3: Yes

4. Is the manuscript presented in an intelligible fashion and written in standard English?

Reviewer #1: No

Reviewer #2: Yes

Reviewer #3: Yes

Reviewer #1: The manuscript explores the genetic polymorphisms within key vitamin D metabolism genes and their association with multiple sclerosis (MS) in the Jordanian population. The topic is not new and several Authors in several populations explored the influence vitamin D related genetic polymorphisms in MS. Additionally, The use of PCR-RFLP is acceptable for genotyping, but it is an older method. Justify the choice of PCR-RFLP versus other high-throughput methods (e.g., TaqMan, sequencing).

Include details about genotyping error rate, call rates, and how ambiguous genotypes were handled. A few SNPs (e.g., rs12794714, rs10877012) are not in HWE in the control group, which could suggest genotyping errors, population stratification, or selection bias. Please, Investigate and explain these deviations. Consider reanalyzing the results with these SNPs excluded or conduct sensitivity analysis. Multiple comparisons are well-handled with Bonferroni correction (though conservative), but some associations remain marginal (e.g., rs4588, p = 0.046).

The power analysis is vague. The authors mention a formula but do not show the actual parameters used or how the final sample size was determined. Consider using FDR (false discovery rate) instead of Bonferroni for less conservative correction. The manuscript presents the results clearly but sometimes overstates significance, especially for results with marginal p-values or weak odds ratios (e.g., OR = 0.35 for rs4588). Temper conclusions where statistical evidence is limited or borderline.

Include effect sizes and confidence intervals consistently for all SNPs. The discussion is exhaustive but needs more focused synthesis rather than listing contradictory studies. Group similar findings thematically (e.g., CYP27B1 findings in Arab vs. European populations).

Use meta-analyses or systematic reviews to contextualize where the findings fall within the broader evidence base.

The manuscript is generally understandable, but some sections are repetitive, and there are numerous grammatical errors, awkward phrasing, and redundant statements. I suggest to use professional editing or language polishing services.

Reviewer #2: It is a well written article with novel in investigate one of the most important subjects in Jordan. Also, an informative presentation of the results. However, I have the following comment for the authors

- You need to add a paragraph about your research limits.

Reviewer #3: The manuscript is well-structured and presents valuable insights into the genetic associations between vitamin D metabolism-related gene polymorphisms and multiple sclerosis in the Jordanian population. However, several minor issues should be addressed to improve clarity and presentation.

Abstract

Abbreviations are not mentioned throughout the manuscript.

Introduction:

Clinical manifestations of multiple sclerosis (MS) are not monitored clearly as a general then selected on MS.

Some sentences in brief about vitamins in general are not mentioned.

There are many types of vitamin D, why select 2 and 3 only?

Abbreviation of vit amin D is Vit D instead of VD.

Line 46 What is meant by B- rays?

Line 48 What is meant by p 450?

Line 64 Diabetes type 1 was written by different manner.

Results

Discussion

English editing is required.

Line 285 Abbreviation is after the word.

Line 289 25OH is meant what?

Line 324 all the paragraph about vit D3, What about vit D2?

References

References to supplementary figures and tables should be checked to ensure they are provided and properly labelled.

Conclusion

What about limitations of this study?

**Do you want your identity to be public for this peer review?** For information about this choice, including consent withdrawal, please see our Privacy Policy

Reviewer #1: No

Reviewer #2: No

Reviewer #3: No

---

## [Author Response · Author response to Decision Letter 1]

11 Sep 2025

Dear Editor in Chief,

I would like to extend my deepest thanks to the reviewers for their constructive comments and suggestions about the manuscript titled “Investigating Genetic Links of Vitamin D Metabolism Pathway Genes (CYP2R1, CYP27B1, CYP24A1, and DBP) in Multiple Sclerosis Patients”. I am pleased to submit the revised version of the paper, which addresses each of the reviewers’ comments. An English language editor also reviewed the manuscript to enhance its flow and the communicability of its scientific content.

Reviewer 1:

The manuscript explores the genetic polymorphisms within key vitamin D metabolism genes and their association with multiple sclerosis (MS) in the Jordanian population.

The topic is not new and several Authors in several populations explored the influence vitamin D related genetic polymorphisms in MS.

We sincerely thank the reviewer for the detailed and constructive feedback. Your thorough observations addressed almost every aspect of the manuscript and greatly helped us identify key areas for improvement. We carefully responded to each point raised, including statistical analysis, data interpretation, methodological clarification, and language editing. Your input has significantly enhanced the quality and clarity of our work.

Although several studies have investigated the relationship between polymorphisms in vitamin D metabolism genes and multiple sclerosis (MS) in various populations worldwide, our study is the first to focus specifically on the Jordanian community. There is a notable scarcity of such comprehensive research within Arab populations compared to European or Latin American communities, where numerous studies have demonstrated significant associations. Therefore, our research fills a crucial gap in the scientific literature, providing essential insights into the genetic factors influencing MS risk within our population. This highlights the importance of conducting further studies in Arab and regional contexts to understand better the role of these genetic variants, which can ultimately inform the development of tailored preventive and therapeutic strategies within our communities.

Additionally, The use of PCR-RFLP is acceptable for genotyping, but it is an older method. Justify the choice of PCR-RFLP versus other high-throughput methods (e.g., TaqMan, sequencing).

The study was carried out from 2020 to 2022, a time when access to high-throughput genotyping platforms like TaqMan or next-generation sequencing was limited due to resource constraints and equipment availability. Under these conditions, PCR-RFLP remained a practical and reliable method for genotyping, especially when paired with strict protocols to ensure accuracy and reproducibility. Although PCR-RFLP is considered a traditional technique, it is a well-established, cost-effective, and accurate method suited to the scope of our research. We emphasized rigorous quality control measures to reduce potential errors, ensuring the validity of our genotyping results despite using this conventional approach.

Include details about genotyping error rate, call rates, and how ambiguous genotypes were handled.

We thank the reviewer for this valuable suggestion. In response, we have added a detailed description of the genotyping quality control procedures. Specifically, we now report the genotyping call rates for both SNPs and samples, the error rate as assessed using duplicate samples, and the criteria used to define and exclude ambiguous genotypes. These details have been incorporated into the “SNP Selection and Genotyping” section to clarify the reliability and integrity of the genotyping data.

A few SNPs (e.g., rs12794714, rs10877012) are not in HWE in the control group, which could suggest genotyping errors, population stratification, or selection bias.

- Please, Investigate and explain these deviations.

- Consider reanalyzing the results with these SNPs excluded or conduct sensitivity analysis.

We recognize that the genetic polymorphism rs10877012 deviates from the Hardy–Weinberg Equilibrium (HWE) in the control group. However, excluding this SNP solely because of this deviation could lead to missing a potentially important genetic variant related to the disease being studied. Several studies have warned against automatically excluding SNPs that deviate from HWE, especially when they show statistically significant associations, as observed in our study.

For instance, (Wigginton, Cutler, and Abecasis 2005; Hosking et al. 2004) noted that deviations from HWE can arise from population structure or genuine biological effects on allele distribution rather than solely from technical errors. Similarly, a study published in the European Journal of Human Genetics (Fardo et al. 2009) emphasized that some SNPs deviating from HWE may be genuinely associated with disease risk, and their exclusion could result in the dismissal of true biological associations.

(Zintzaras 2010) further emphasized in the European Journal of Epidemiology that "strictly enforcing HWE in control groups may lead to the loss of valuable information regarding genotype-disease relationships," especially in the context of complex diseases or selective pressures.

Therefore, due to the strong and consistent associations observed with rs10877012, we chose to include it in the primary analysis. At the same time, we conducted a sensitivity analysis excluding this SNP, which confirmed the stability and robustness of the overall results, in line with prior research recommendations. Additionally, other SNPs (e.g., rs12794714, rs7041, and rs4588) showed slight deviations from Hardy–Weinberg Equilibrium in the control group, with p-values ranging from 0.02 to 0.04. These deviations were minor and common for moderate-sized population studies. As supported by previous research, such small deviations may stem from random variation, limited sample size, or subtle population structure rather than genotyping errors. Importantly, these SNPs demonstrated consistent and statistically significant associations with multiple sclerosis across various genetic models. Therefore, we included them in our main analysis while transparently reporting their HWE values in the results section.

Finally, we followed strict protocols when selecting study participants. Samples from related individuals or those with unclear demographic or clinical data were excluded to prevent potential confounding factors. While this careful selection was crucial for maintaining data integrity, it also reduced the overall sample size. Furthermore, due to the conservative nature of the target population and cultural constraints, collecting larger, more genetically diverse samples was difficult. These limitations may have contributed to the slight deviations from Hardy–Weinberg equilibrium (HWE) seen in a few single-nucleotide polymorphisms (SNPs) and the limited allele diversity at some loci.

References

Fardo, David W, K David Becker, Lars Bertram, Rudolph E Tanzi, and Christoph Lange. 2009. “Recovering Unused Information in Genome-Wide Association Studies: The Benefit of Analyzing SNPs out of Hardy–Weinberg Equilibrium.” European Journal of Human Genetics 17 (12): 1676–82. https://doi.org/10.1038/ejhg.2009.85.

Hosking, Louise, Sheena Lumsden, Karen Lewis, Astrid Yeo, Linda McCarthy, Aruna Bansal, John Riley, Ian Purvis, and Chun-Fang Xu. 2004. “Detection of Genotyping Errors by Hardy–Weinberg Equilibrium Testing.” European Journal of Human Genetics 12 (5): 395–99. https://doi.org/10.1038/sj.ejhg.5201164.

Wigginton, Janis E., David J. Cutler, and Gonçalo R. Abecasis. 2005. “A Note on Exact Tests of Hardy-Weinberg Equilibrium.” The American Journal of Human Genetics 76 (5): 887–93. https://doi.org/10.1086/429864.

Zintzaras, Elias. 2010. “Impact of Hardy–Weinberg Equilibrium Deviation on Allele-Based Risk Effect of Genetic Association Studies and Meta-Analysis.” European Journal of Epidemiology 25 (8): 553–60. https://doi.org/10.1007/s10654-010-9467-z.

Multiple comparisons are well-handled with Bonferroni correction (though conservative), but some associations remain marginal (e.g., rs4588, p = 0.046).

We want to clarify that our use of the Bonferroni correction was based on well-established statistical principles. When performing multiple tests, it is a conservative and widely accepted method to reduce the rate of false positives (Type I errors) when testing multiple hypotheses simultaneously. In our study, the significance threshold (α = 0.05) was divided by the number of independent comparisons to ensure that the results are truly statistically significant while decreasing the likelihood of false associations caused by multiple testing. This approach is explained in the Results section and noted in the footnote of each relevant table. Although the Bonferroni correction may slightly lower statistical power, it increases the reliability of the findings and is commonly used in genetic studies involving multiple SNPs.

We emphasize that applying this method demonstrates our adherence to proper scientific practices and that our results remain statistically significant after correction, confirming their reliability despite the statistical challenges of analyzing genetic data.

The power analysis is vague. The authors mention a formula but do not show the actual parameters used or how the final sample size was determined.

Thank you for your comment. We recognize that the previous power analysis description was limited, and it did not include all parameters and calculation details. Because the original calculation using OpenEpi’s “Frequency in a Population” option is more suitable for prevalence estimation than for case–control genetic studies, we have removed that calculation from the Methods section. Instead, we now mention this as a study limitation, noting that the sample size may have been too small to detect small to moderate genetic effects.

Consider using FDR (false discovery rate) instead of Bonferroni for less conservative correction.

Thank you for the suggestion. Our analyses were hypothesis-driven with a small number of predefined SNP comparisons. For this reason, we used the Bonferroni correction, which provides a conservative and appropriate control of type I error. While FDR is useful for large-scale or exploratory analyses, it was not necessary in our study, as Bonferroni correction adequately accounted for multiple comparisons. We will, however, consider applying FDR in future studies involving more comparisons or more exploratory designs.

The manuscript presents the results clearly but sometimes overstates significance, especially for results with marginal p-values or weak odds ratios (e.g., OR = 0.35 for rs4588).

➤ Temper conclusions where statistical evidence is limited or borderline.

We agree on the importance of cautious interpretation of results, especially those with marginal p-values or odds ratios near 1. For example, the finding related to rs4588 (OR = 0.35, p = 0.042) should be viewed with caution, and our conclusions are adjusted accordingly to avoid overstating the statistical significance of borderline results.

Include effect sizes and confidence intervals consistently for all SNPs.

Thank you for your comment. We have thoroughly ensured that effect sizes, expressed as odds ratios, along with their 95% confidence intervals, are consistently reported for all SNPs across every tested genetic model. This information is included in Tables 6 to ensure a comprehensive and transparent presentation of the associations.

The discussion is exhaustive but needs more focused synthesis rather than listing contradictory studies.

Group similar findings thematically (e.g., CYP27B1 findings in Arab vs. European populations).

Use meta-analyses or systematic reviews to contextualize where the findings fall within the broader evidence base.

Thank you for this helpful suggestion. We agree that organizing the findings by theme can improve the discussion. In the revised version, we have restructured the discussion to highlight patterns within different population groups where possible (e.g., Arab versus European cohorts). However, we also recognize that the number of studies involving Arab populations remains limited, which restricts the depth of regional comparisons. Similarly, only a few meta-analyses on these polymorphisms in relation to MS are available, which limits our ability to place the findings in a broader context. Nonetheless, we have cited and integrated the relevant systematic reviews, while emphasizing the need for more large-scale studies and meta-analyses, especially in underrepresented populations.

Reviewer 2:

It is a well written article with novel in investigate one of the most important subjects in Jordan. Also, an informative presentation of the results. However, I have the following comment for the authors

- You need to add a paragraph about your research limits.

We sincerely appreciate your valuable comment and positive evaluation of our manuscript. In response to your suggestion, we have added a dedicated paragraph discussing the limitations of our study at the end of the conclusion section. This new section outlines the methodological and contextual constraints of our work, including considerations of sample size, potential population stratification, and the limitations of the genotyping technology employed. We believe this addition enhances the transparency and scientific rigor of our manuscript.

Reviewer 3:

The manuscript is well-structured and presents valuable insights into the genetic associations between vitamin D metabolism-related gene polymorphisms and multiple sclerosis in the Jordanian population. However, several minor issues should be addressed to improve clarity and presentation.

Abstract

Abbreviations are not mentioned throughout the manuscript.

We sincerely appreciate your thoughtful and precise comments. Your focus on both content and presentation helped us enhance the clarity, consistency, and organization of the manuscript. We have incorporated your suggestions regarding terminology and formatting. Thank you for your valuable input. We also revised the abstract and made sure all abbreviations are clearly defined at first mention.

Introduction:

The clinical manifestations of multiple sclerosis (MS) are not clearly monitored or selected in relation to MS. Some brief sentences about vitamins in general are not included.

We have reorganized the introduction to better explain the clinical signs of MS and give a short background on the overall role of vitamins.

There are many types of vitamin D, why select 2 and 3 only?

We focused on vitamin D2 and D3 because they are the two biologically relevant forms of vitamin D in humans. Both are primarily converted in the circulation and are used to evaluate vitamin D status. Other forms, such as D4 or D5, are either biologically inactive or not clinically significant.

Abbreviation of vitamin D is Vit D instead of VD.

We appreciate the reviewer’s comment. We have carefully reviewed the manuscript and corrected the terminology throughout by replacing all incorrect instances of 'VD' with the proper abbreviation, Vit D.

Line 46 What is meant by B- rays?

Line 48 What is meant by p 450?

Line 64 Diabetes type 1 was written by different manner.

We appreciate the reviewer for the helpful comments. The term "B-rays" has been clarified to "ultraviolet B (UVB) rays" for accuracy. The reference to "p 450" has been corrected to "cytochrome P450" to match the proper terminology. Additionally, "Diabetes type 1" has been revised to the standard format "Type 1 diabetes" for consistency throughout the manuscript.

Results

Discussion

English editing is required.

We have thoroughly revised the manuscript to enhance the English language and overall clarity.

Line 285 Abbreviation is after the word.

The placement of the abbreviation has been adjusted to appear right after the full term in parentheses, following standard scientific writing practices.

Line 289 25OH is meant what?

We clarified the term by replacing “25OH” with the correct form “25(OH)D” in the revised manuscript to ensure clarity and accuracy.

Line 324 all the paragraph about vit D3, What about vit D2?

We clarified in the revised manuscript that our discussion

---

## [Editor Report · Decision Letter 1]

22 Sep 2025

Investigating Genetic Links of Vitamin D Metabolism Pathway Genes (CYP2R1, CYP27B1, CYP24A1, and DBP) in Multiple Sclerosis Patients

PONE-D-25-04705R1

Dear Dr. Al-Eitan,

We’re pleased to inform you that your manuscript has been judged scientifically suitable for publication and will be formally accepted for publication once it meets all outstanding technical requirements.

Kind regards,

Wajdy J. Al-Awaida, Ph.D

Academic Editor

PLOS ONE
---

## [Editor Report · Acceptance letter]

PONE-D-25-04705R1

PLOS ONE

Dear Dr. Al-Eitan,

I'm pleased to inform you that your manuscript has been deemed suitable for publication in PLOS ONE. Congratulations! Your manuscript is now being handed over to our production team.

Kind regards,

on behalf of

Prof. Wajdy J. Al-Awaida

Academic Editor

PLOS ONE